# Effect of UV Radiation and Salt Stress on the Accumulation of Economically Relevant Secondary Metabolites in Bell Pepper Plants

**Jan Ellenberger * , Nils Siefen, Priska Krefting, Jan-Bernd Schulze Lutum, Daniel Pfarr, Maja Remmel, Lukas Schröder and Simone Röhlen-Schmittgen**

INRES Horticultural Sciences, University of Bonn, Auf dem Huegel 6, 53121 Bonn, Germany; nils.siefen@googlemail.com (N.S.); priska-krefting@gmx.de (P.K.); jbsl@uni-bonn.de (J.-B.S.L.); daniel-pfarr@gmx.de (D.P.); s7maremm@uni-bonn.de (M.R.); lukas-schroeder@onlinehom.de (L.S.); s.schmittgen@uni-bonn.de (S.R.-S.)

* Correspondence: ellenberger@uni-bonn.de

**Abstract:** The green biomass of horticultural plants contains valuable secondary metabolites (SM), which can potentially be extracted and sold. When exposed to stress, plants accumulate higher amounts of these SMs, making the extraction and commercialization even more attractive. We evaluated the potential for accumulating the flavones cynaroside and graveobioside A in leaves of two bell pepper cultivars (Mavras and Stayer) when exposed to salt stress (100 mM NaCl), UVA/B excitation (UVA 4–5 W/m$^2$; UVB 10–14 W/m$^2$ for 3 h per day), or a combination of both stressors. Plant age during the trials was 32–48 days. HPLC analyses proved the enhanced accumulation of both metabolites under stress conditions. Cynaroside accumulation is effectively triggered by high-UV stress, whereas graveobioside A contents increase under salt stress. Highest contents of secondary metabolites were observed in plants exposed to combined stress. Effects of stress on overall plant performance differed significantly between treatments, with least negative impact on above ground biomass found for high-UV stressed plants. The usage of two non-destructive instruments (Dualex and Multiplex) allowed us to gain insights into the ontogenetical effects at the leaf level and temporal development of SM contents. Indices provided by those devices correlate fairly with amounts detected via HPLC (Cynaroside: $r^2$ = 0.46–0.66; Graveobioside A: $r^2$ = 0.51–0.71). The concentrations of both metabolites tend to decrease at leaf level during the ontogenetical development even under stress conditions. High-UV stress should be considered as a tool for enriching plant leaves with valuable SM. Effects on the performance of plants throughout a complete production cycle should be evaluated in future trials. All data is available online.

**Keywords:** *Capsicum annuum*; flavonoids; fluorescence monitoring; bio-waste utilization

---

## 1. Introduction

### 1.1. Green Biomass as a Source of Valuable Chemicals

Commercial vegetable production is accompanied by large quantities of so far under-utilized green biomass in all stages of production and especially after harvest [1]. While the use of biomass for the purpose of energy production is becoming a standard procedure in northern Europe in recent years [2], the extraction and the use of high-value secondary metabolites (SMs) from vegetable plant leaves are just being developed. Research strategies in Europe are heading toward a cascade use of agricultural byproducts and pave the way for extracting and using "valuable substances or molecules before ultimately discarding the left-overs" [3]. The pharmaceutical industry—as an example—is

highly dependent on plant SMs, since approximately 60% of anticancer compounds and 75% of drugs for infectious diseases are derived from plants [4]. In this frame, research on targeted enrichment of valuable substances in plant biomass is gaining importance [5].

### 1.2. Plant Stress as a Measure to Increase Leaf Secondary Metabolite Content

The biochemical background of enhanced accumulation of SMs in plant leaves as a measure to cope with stress is a well-described phenomenon [2,6,7]. In short, the cultivation of plants under suboptimal conditions leads to an increased amount of reactive oxygen species (ROS) in plant tissues. Accumulation of SMs is a plant strategy to avoid oxidative damage caused by reactive oxygen species [8]. In theory, both biotic and abiotic stressors could lead to higher amounts of valuable SMs in plants. While biotic stressors such as fungi and insects are hard to control and may cause major phytosanitary problems, abiotic stressors are easier to manage and applicable by practitioners. The results of several studies in recent years indicate that abiotic stressors are a useful tool for SM accumulation in leaves of horticultural plants. Secondary metabolites in *Centella asiatica* leaves increase under enhanced UV-B light, especially in the epidermis [9]. In bell pepper, increased flavonoid contents can be found in leaves exposed to elevated UV [10]. The promoting effect of UV-B radiation on flavonoid accumulation in plant leaves has recently been reviewed [11]. The effects of salt stress on the antioxidant machinery may be adverse and depend on the plant's tolerance [12] and salt concentration in the rootzone [13]. Another extensive study on leaf metabolism in bell pepper under different levels of salt stress revealed an increasing reduction in growth with increasing NaCl contents in the rootzone [14]. While tolerant plants increase leaf SM contents to cope with salt stress, sensitive plants do not have this mechanism and senesce, finally dying off if the stressor is persistent [12]. Studies directly comparing effects of salt and UV stress on leaf SMs are rare. One study shows both stressors to similarly affect leaf contents of the flavonoids quercetin and luteolin in *Ligustrum vulgare* [15]. Abiotic stressors such as drought and salt stress are easily applicable in commercial greenhouse production in soilless systems, which are the predominant systems in many parts of the world, including Europe [16].

### 1.3. Non-Invasive Monitoring of Secondary Metabolites in Plant Leaves

Quantification of secondary metabolites including flavonoids with portable optical devices is well established in plant sciences [17]. The use of non-invasive optical sensors to investigate plant leaf components has several advantages over laboratory analyses: data acquisition is faster and more cost effective than laboratory analyses [18]. Moreover, considerate handling of leaves allows for several measurements of the same leaf, enabling to gain insights in temporal developments. Several studies demonstrated the viability of optical devices to access secondary metabolites in plant leaves: a multiparametric fluorescence sensor was used to evaluate the influence of nutrient deficiency on the chemical properties of tomato leaves and to quantify the content of the flavonoids rutin and solanesol [19,20]. In bell pepper, a fluorescence sensor was used to evaluate the impact of priming plants with high light conditions on leaf flavonoid content [10].

### 1.4. Cynaroside and Graveobioside A

The vast diversity and chemical complexity of plant SMs often prohibit an economically feasible chemical synthesis. Therefore, extraction either from wild or cultivated plants often represents the best source of supply [1].

Cynaroside (Luteolin-7-glucoside) potentially has a range of medicinal applications: it has the capability to prevent ROS-induced apoptosis in heart cells [21]. Cynaroside furthermore diminishes kidney injury as a side effect of cancer treatments with the chemotherapeutic drug cisplatin. A potential medicinal use of graveobioside A (Luteolin-7-apiosyl-glucoside) is proven by a patent on its application in preparation of drugs for preventing hyperuricemia and gout [22]. Graveobioside A was shown to be contained in several plants, such as celery seeds, parsley, and bell pepper [23,24].

Several SMs in Solanaceae leaves have the potential to biologically control insects [25]. Graveobioside A is such a potential natural insecticide, since oviposition of the American serpentine leafminer fly (*Liriomyza trifolii*) was shown to drop in kidney bean leaves treated with a graveobioside A containing solution [24]. It is expected that the demand for natural insecticides will increase across the EU due to more rigid legislation [26].

We hypothesize that cynaroside and graveobioside A contents in bell pepper leaves can be enhanced by abiotic stressors that are potentially applicable by practitioners in the future. Another aim is to check whether non-invasive devices can be used for assessments of cynaroside and graveobioside A in bell pepper leaves. Furthermore, we attempt to get insights in interactions between different stressors and differences in stress response between two bell pepper cultivars.

## 2. Material and Methods

### 2.1. Plant Material and Growth Conditions

Seeds of sweet pepper plants (*Capsicum annuum*) cultivar 'Stayer' (Rijk Zwaan, De Lier, The Netherands) and 'Mavras' (Enza Zaden, Enkhuizen, The Netherlands) were sown in soil under greenhouse conditions. Fourteen-days old pepper plants were transplanted into small rockwool cubes ($3 \times 3 \times 5$ cm) and one further week later into larger cubes ($10 \times 10 \times 7.5$ cm) (Grotop Master, Grodan, The Netherlands). On day 32 after seeding, plants were transferred to a grow chamber to ensure a stable environment. From that day on, stress was applied for 16 days, resulting in a plant age of 48 days at the end of the trial. A longer trial was not feasible due to limitations of the chosen facility. All plants received all nutrients mandatory for optimal growth prepared from two stock solutions (17.2 mM nitrogen, 5.4 mM calcium, 4.7 mM potassium, 0.4 mM phosphorous, 5.4 mM sulfur, 2.4 mM magnesium, 0.01 mM iron and micronutrients; electrical conductivity 2.5 mS cm$^{-1}$; pH 5.5). Plants were cultivated at the greenhouse facility in Bonn-Endenich (University of Bonn, Bonn, Germany) at day/night temperatures of 24.5 °C ± 5.4 and supplemental light intensity of 203–540 μm m$^{-2}$s$^{-1}$ provided by sodium vapor lamps (Philips Lighting GmbH, Hamburg, Germany).

To apply salt stress, a salt concentration of 100 mM NaCl for a period of 16 days was added to the standard nutrient solution, since that concentration was shown to trigger a higher total phenolic content in leaves of bell pepper seedlings in a previous study [14]. To apply UV stress, plants were exposed to UV light (UVA 4–5 W m$^{-2}$; UVB 10–14 W m$^{-2}$) for 3 h per day (Philips Lighting GmbH, Hamburg, Germany) over a 16-day period. In addition, some plants were exposed to combined salt and UV stress. Plant age at stress onset was 32 days. A total of 5 plants per treatment (control, salt stress, UV stress, combined stress) were randomized in the growth chamber.

### 2.2. Non-Destructive Recordings

Non-destructive measurements were performed on all leaves per plant, from mature to young. Measurements were conducted using two well-established devices in stress physiology monitoring. First device is the multiparametric fluorescence excitation system Multiplex® (Multiplex®, Force-A, Orsay, France), described in previous studies [27]. All recordings with this device were done at a constant distance of 0.10 m to the leaf surface and a frontal cover plate with an aperture of 4 cm in diameter opening to assess the index of epidermal flavonols (FLAV index): $\log \frac{FRF\_R}{FRF\_UV}$.

Secondly, the transmittance-based fluorescence measurements were conducted with the Dualex sensor (Force-A, Orsay, France). The Dualex is a device with a leaf-clip; measurements were taken with virtually no distance to the leaf surface. The device is extensively described in the literature [28,29].

### 2.3. Plant Harvest

Plants were harvested 16 days after treatment inception (DATI) at a plant age of 48 days. The total fresh weight of shoots was determined immediately. Leaves were dried for 7 days at 50 °C (drying oven) to collect dry weights.

## 2.4. Leaf Sample Preparation and Laboratory Analysis

Samples were taken at the harvesting at 16 DATI, of the mature leaf 4 and the young leaves 10 and 12, to assess the impact of stress application on the amount of the two luteolins, graveobioside A and cynaroside. All leaf numbers are given as the number of true leaves, counted from the base of the plant. The samples were freeze-dried and then stored at −20 °C until further processing. Ground leaf samples were prepared for HPLC determination (Agilent 1260 Infinity HPLC System Agilent Technology Deutschland GmbH, Ratingen, Germany). An amount of 0.3 g was extracted with water-diluted methanol (60:40, *v/v*) for 10 min in an ultrasonic bath, centrifuged for 10 min at 4 °C with 13,000 rpm (Centrifuge 5415R, Eppendorf AG, Hamburg, Deutschland) repeated four times. The supernatants were collected and stored at −20 °C until HPLC analysis. The samples were filtrated through a membrane filter (Phenomenex, Aschaffenburg, Germany) prior to injection. The HPLC system consisted of an autosampler, a diode array UV–Vis detector and was coupled with a quaternary solvent delivery system. The column (Nocleodur C18, $3 \times 150$ mm, 3 µm, Macherey-Nagel, GmbH & Co. KG, Düren, Germany) was isocratically eluted with a binary mixture of water and methanol (60:40) adjusted to pH 2.8 with phosphoric acid. The flow rate was 0.3 mL min$^{-1}$; 10 µL samples were injected onto the column equilibrated at 25 °C (detection at 355 nm). Graveobioside A peak was detected at 14.1 min, and cynaroside at 15.6 min. Both calibration curves were obtained from diluted series of standards provided by PhytoLab (Vestenbergsgreuth, Germany).

## 2.5. Data Analysis and Statistics

All data is available online [30]. Data analysis was performed with R (R Core Team, Vienna, Austria) [31] in RStudio (R Studio Team, Boston, USA) [32]. According to the data structure, e.g., balanced or imbalanced, type I or type III ANOVA were used to compare group means. Applied post-hoc test was Tukey's HSD. Figures were created in RStudio, with the package ggplot2 [33].

## 3. Results

### 3.1. Stress-Related Effect Varies Among Secondary Metabolites and Cultivars

A treatment effect was observed on contents of both cynaroside and graveobioside A, while no significant effect of the variable cultivar on either metabolite content was found. There was a strong tendency for higher graveobioside A in 'Mavras' as compared to Stayer ($p = 0.055$). No interactions between cultivar and treatment were observed (Table 1). Both combined-stressed plants and plants under UV-exposure accumulated significantly higher amounts of cynaroside in their leaves than control and salt-stressed plants (Figure 1, A + B). Plants of the cultivar 'Mavras' accumulated significantly higher graveobioside A amounts in salt-stressed and combined-stressed plants than in control and UV-stressed plants (Figure 1C). No significant treatment effect on graveobioside A content in plants of the cultivar Stayer was found (Figure 1D). Levels of SM in leaves of different ontogenetical stages are shown as an illustration of uneven distribution within the plants. SM contents decrease with leaf ontogenetical stage (Figure 1).

**Table 1.** Interaction and main effect for treatments (control, salt-stress, combined-stress, UV-stress) and cultivars (Mavras and Stayer), calculated with a type I two-way ANOVA. Grayed area indicates significant effect ($p \leq 0.001$).

| Factor | Cultivar | Treatment | Cultivar × Treatment |
|---|---|---|---|
| Cynaroside | 0.179 | $<2 \times 10^{-16}$ | 0.917 |
| Graveobioside A | 0.055 | $1.25 \times 10^{-5}$ | 0.141 |
| Dry Weight | 0.00082 | $3.8 \times 10^{-12}$ | 0.426 |
| Fresh Weight | 0.00017 | $1.15 \times 10^{-15}$ | 0.146 |

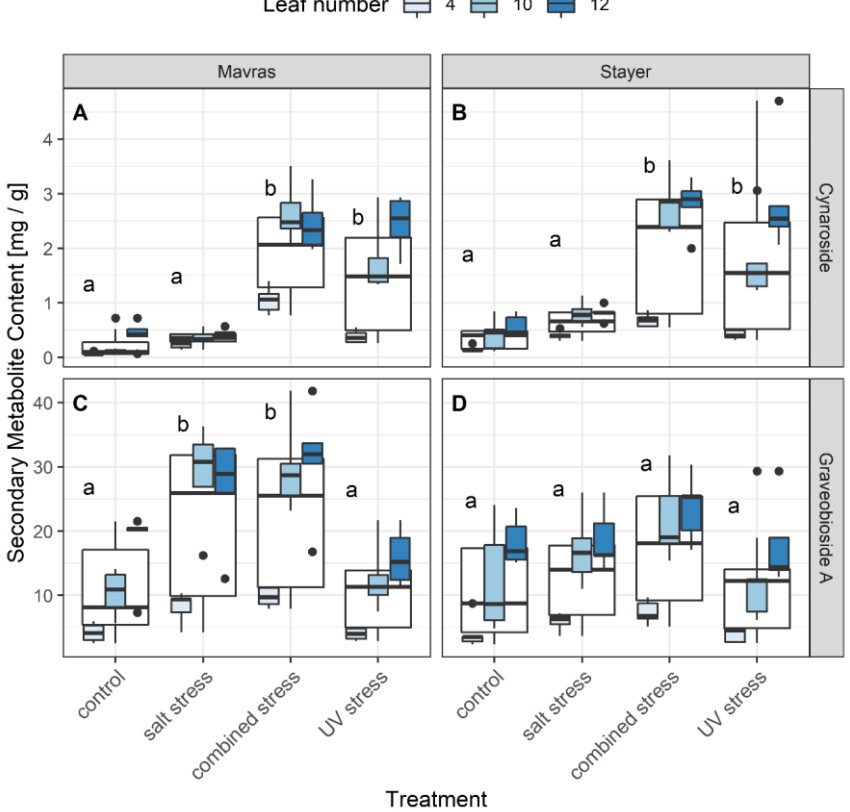

**Figure 1.** HPLC-determined leaf cynaroside (**A**,**B**) and graveobioside A (**C**,**D**) contents, for bell pepper cultivars 'Mavras' (**A**,**C**) and 'Stayer' (**B**,**D**) under different growth conditions, 15 days after treatment inception ($n = 5$). Transparent boxplots show pooled data from all leaves ($n = 15$). Colored boxplots represent leaf age—subgroups (Leaf 4, 10, and 12 as counted from the base, with darkest colors for youngest leaves). Letters (a,b) indicate differences within each cultivar × secondary metabolite—combination (Tukey HSD, $p < 0.05$).

Both fresh and dry weight of bell pepper plants differed significantly depending on the cultivar, with Stayer attaining higher weights than Mavras. Treatment had a significant effect on both fresh and dry weight. There was no interaction between the treatment and cultivar regarding plant's fresh or dry weight. Dry weight of plants of the cultivar Mavras was significantly higher in control plants than in any other treatment (Table 1). UV-stressed plants of both tested cultivars exhibited higher fresh and dry weights than plants under salt-stress and combined-stress conditions (Figure 2). Observed mean fresh weight decreased in salt-stress and combined-stress plants compared to control and UV stress, which were in the magnitude of 50% (Figure 2C,D). The mean dry weight tended to be higher for salt-stressed plants as compared to plants under combined stress, but lower than the dry weights of plants experiencing UV stress or control conditions (Figure 2A,B).

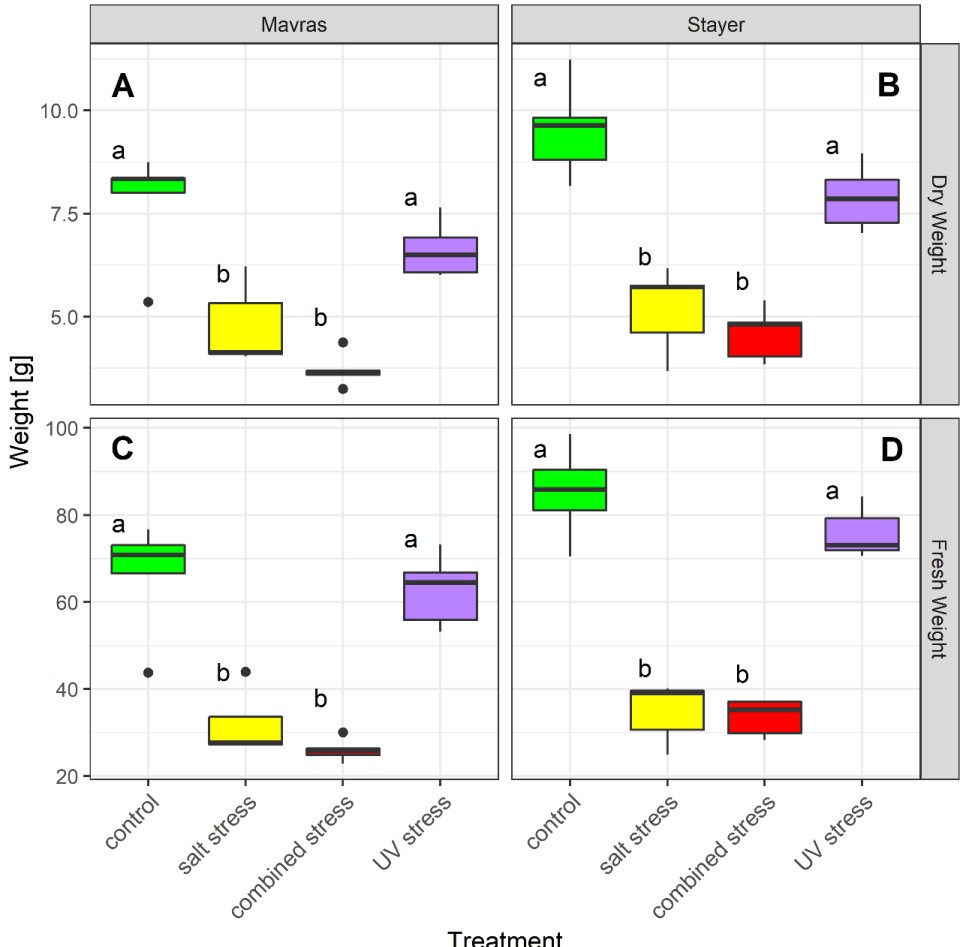

**Figure 2.** Aboveground biomass (dry weight: (**A**), (**B**); fresh weight: (**C**), (**D**)) of bell pepper cultivars "Mavras" (**A**), (**C**) and "Stayer" (**B**), (**D**) under different growth conditions, 15 days after treatment induction (*n* = 5). Letters (a,b) indicate differences within each cultivar × dry/fresh weight—combination (Tukey HSD, *p* < 0.05).

### 3.2. Non-Invasive Monitoring of Secondary Metabolites

Figure 3 shows exponential regressions between three indices (Multiplex indices FLAV and NBI_R; Figure 3A–D and Dualex index Flav; Figure 3E,F) and leaf contents of the SMs cynaroside (Figure 3A,C,E) and graveobioside A (Figure 3B,D,F), respectively. Predictions of graveobioside A contents based on the indices are better than predictions of cynaroside contents. Multiplex indices are more accurate predictors than the Dualex index, as outlined by correlation coefficients ($r^2$). Index values level off at cynaroside contents above 1.5 mg g$^{-1}$. The connection between graveobioside A and the indices is more linear, but still leveling off at graveobioside A contents above approximately 25 mg g$^{-1}$.

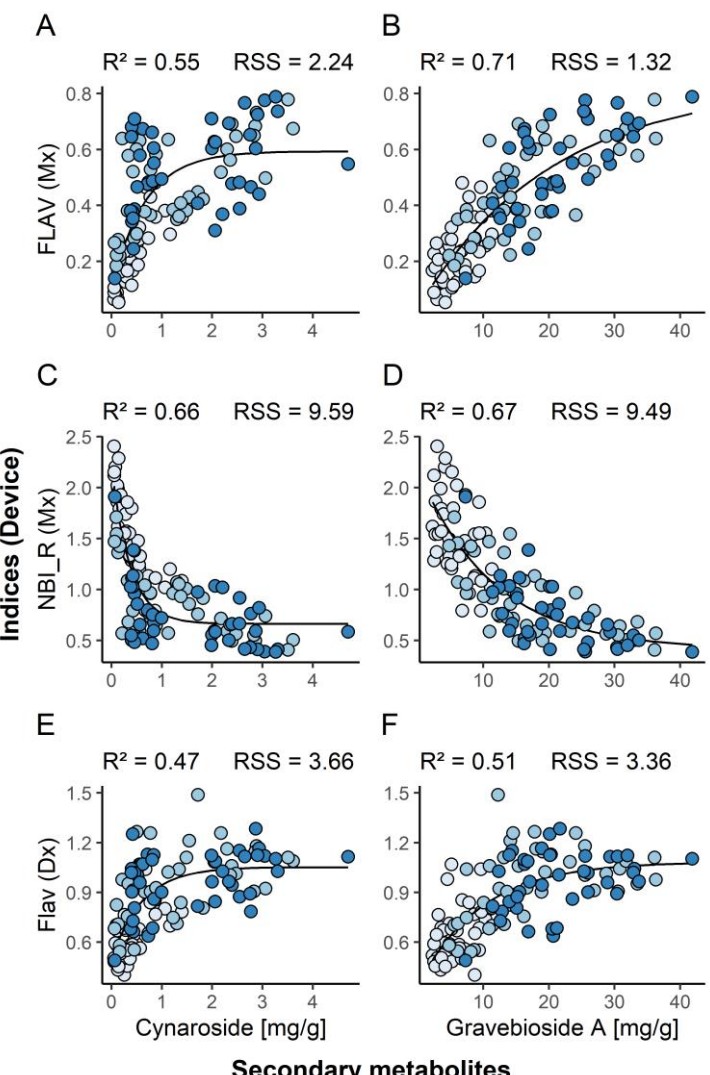

**Figure 3.** Exponential regression between indices of non-invasive devices and leaf secondary metabolite concentrations in bell pepper leaves, determined via HPLC. Contents of cynaroside and graveobioside A correlated with FLAV (Mx) (**A**), (**B**), NBI_R (Mx) (**C**), (**D**), and Flav (Dx) (**E**), (**F**). Color of points represents leaf age (Leaf 4, 10, and 12 as counted from the base, with darkest colors for youngest leaves). Lines indicate exponential regressions (*n* = 60). RSS, residual sum of squares.

### 3.3. Spatial and Temporal Development of Secondary Metabolite Contents

The only significant changes in FLAV values within cultivar × treatment groups were seen among the fourth leaves of combined-stressed Mavras plants at days 0 versus 9 and 0 versus 15, respectively (Figure 4C). A clear trend was observed for the fourth leaves of combined-stressed Stayer plants at days 0 versus 15 (TukeyHSD, *p* = 0.053) (Figure 4D). Generally, FLAV values for stressed plants tend to increase, while the values for control leaves tend to decrease. A comprehensive overview of associated main effects is given in Table 2.

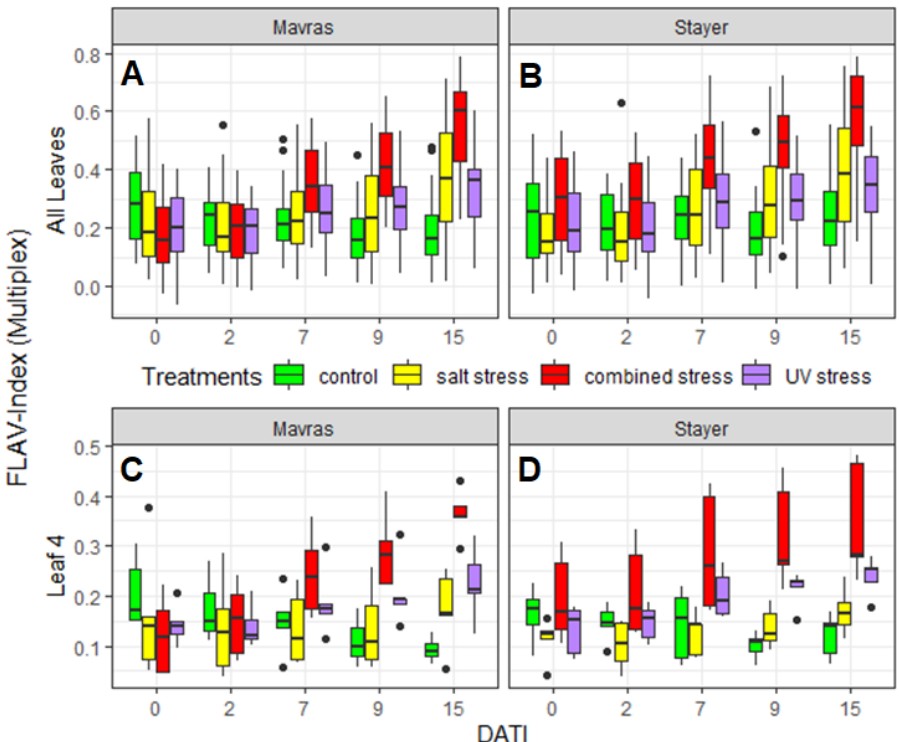

**Figure 4.** Temporal development of secondary metabolites in leaves of bell pepper cultivars "Mavras" and "Stayer", expressed with the FLAV-index (Multiplex). (**C**), (**D**), $n = 5$; (**A**), (**B**), $n = 5\text{–}50$; DATI, day after treatment initiation.

**Table 2.** Interaction and main effect for treatments (control, salt-stress, combined-stress, UV-stress) and DATI (0, 2, 7, 9, 15). To account for the unbalanced design (e.g., unequal numbers of observations within each level of DATI), type III ANOVA was selected to compare differences between factor means for FLAV values of "All leaves". Grayed area indicates significant effect at $p \leq 0.05$ (light), $p \leq 0.01$ (medium), and $p \leq 0.001$ (dark).

| Leaf | Cultivar | Treatment | DATI | Treatment × DATI |
|------|----------|-----------|------|------------------|
| All leaves | Mavras | 0.085 | 0.027 | $<2 \times 10^{-16}$ |
| | Stayer | 0.079 | 0.509 | $2.17 \times 10^{-6}$ |
| Leaf 4 | Mavras | 0.00011 | 0.055 | 0.00027 |
| | Stayer | $8.37 \times 10^{-12}$ | 0.00484 | 0.081 |

## 4. Discussion

We are among the first groups accessing the amount of graveobioside A in pepper leaves [4]. For cynaroside, the range of values detected corresponds to the results of other studies [34,35].

### 4.1. Stress-Related Effect Varies According to Secondary Metabolites and Cultivars

Since cynaroside contents under single UV-stress and combined UV- and salt-stress are not significantly different (Figure 1A,B), cynaroside accumulation appears to be triggered mainly by high radiation conditions. Interestingly, and in contrast to cynaroside, graveobioside A accumulation is triggered more effectively by salt stress than by UV-stress, especially in the cultivar Mavras (Figure 1C). This is a surprising result, since biosynthesis of flavonoids is said to be enhanced similarly by UV radiation and salinity [15,36]. On the other hand, some authors report that the regulation of SM production in response to salt stress differs between salt-sensitive (upregulation) and salt-tolerant (downregulation) plants [12]. However, differences in salt-stress tolerance between the cultivars used in this study are not supported by differing plant biomasses (Figure 2). The chemical group of

flavonoids is highly diverse, and metabolic pathways are not entirely understood to date. At this point, it remains unclear how exactly upregulation of cynaroside synthesis under UV stress and upregulation of graveobioside A synthesis under salt stress occurs.

Our results indicate—as expected—that salt-stressed plants acquire a significantly lower biomass than both control plants and UV-stressed plants. Stunted growth is a well-described symptom of severe salt stress in plants [12,37]. If the applied salt concentration would have been lower, negative effects could probably have been avoided to a certain extent, as recently discussed in a review on the potential of seawater use in soilless culture [13]. Reaction of plants to UV-B exposure varies from growth reduction to enhancement, depending on species, cultivar, and stress level [11,38]. Since the overall aim of the stress application is the accumulation of higher amounts of secondary metabolites in the plant's green biomass, it is necessary to consider not only the share of desired metabolite in the plant´s biomass, but also the biomass reduction caused by the treatment. Considering this background, we can state that stressors with minor negative effects on plant biomass accumulation, but major positive effects on contents of desired metabolites in the plant tissues, are necessary to achieve these aims. Finding the perfect trade-off between biomass and fruit yield loss, on the one hand, and SM increase, on the other hand, will be crucial to improve the production system. In our specific setup with two single stressors and one combined stress, with respective levels of stress described above, the single UV stress is most promising, whereas salt stress (100 mM NaCl), although promoting the accumulation of graveobioside A, is less promising as a tool to enhance whole plant SM amounts, due to the decrease in total biomass. Effects on plants grown over a whole season are a matter of ongoing research.

### 4.2. Non-Invasive Monitoring

The indices provided by both optical devices deliver better estimates for leaf graveobioside A contents than for leaf cynaroside contents. That is an expected result, since the amount of graveobioside A as determined via HPLC is up to ten-fold higher than the amount of cynaroside (0–4 versus 2–40 mg $g^{-1}$) and both secondary metabolites share similar optical properties. Any estimate of concentrations based on non-invasive, optical devices will be best for the predominant fraction of a group of metabolites with similar optical properties. By the same token, signals of metabolites that occur in small quantities are more likely to be superimposed by other signals and therefore difficult to quantify. Additional factors known to influence non-invasive assessment of leaf compounds include the concentration of other pigments potentially influencing the measurement [39], leaf thickness [40], and the device used [41].

In our study, the FLAV-index of the Multiplex shows an almost linear response to changes in leaf graveobioside A content (Figure 2B). The same applies for the NBI_R index, which correlates negatively with the actual graveobioside A content. Both indices use the far-red fluorescence of leaves excited with UV-light and normalize that signal for the red fluorescence emitted after excitation with red light [29]. As an enhanced graveobioside A content leads to a stronger absorption of UV light in the leaf epidermis, less radiation penetrates into the mesophyll, which in turn leads to a lower chlorophyll fluorescence. We have to highlight the broad distribution of fluorescence values, though, which prohibits a precise prediction of actual graveobioside A levels on the individual leaf level. The Flav-index of the Dualex is almost indifferent to changes at graveobioside A levels above 25 mg $g^{-1}$.

None of the indices is strongly related to the leaf cynaroside contents quantified by HPLC. Neither the Dualex nor the Multiplex provide any indices that allow to quantify cynaroside contents higher than approximately 1 mg $g^{-1}$ dry weight. An exact evaluation of high levels of this specific SM in bell pepper leaves is therefore not possible with the tested devices. However, the correlations we have identified between the FLAV index and HPLC measurements still allow us to analyze the gradual changes in SM contents as they occur during the prolonged period of stress.

### 4.3. Insights in Spatial and Temporal Accumulation of Secondary Metabolites

The usage of non-invasive phenotyping tools such as the Multiplex and Dualex devices allows to analyze leaf constituents during ontogenesis. The observed drop of the flavonol content in leaves of unstressed plants during ontogenesis (Figure 4C,D) is in line with the theories that (a) the production of phenolics, such as flavonols, is mainly caused by photodamage [42] and (b) that ontogenetically young leaves are, in general, more prone to be affected by high light stress than older leaves, since their photosynthetic apparatus is not yet well developed [43] and the photoprotective cuticula is thinner compared with older leaves [44]. Therefore, young leaves show stress-related reactions in conditions that are neither stressful for older leaves nor for the entire plant. However, the described ontogenetic effects tend to be overcompensated by stress-related effects in all three stress treatments (Figure 4C,D). Thus, flavonol contents of the fourth leaf as measured with the FLAV (Mx) index slightly increased in plants experiencing single stresses, while plants exposed to combined stress showed major increases in leaf flavonol contents (Figure 4C,D).

### 4.4. Implications and Future Challenges

The present study proves that abiotic stresses, in particular, salt stress and UV stress, can enhance the amount of economically valuable SMs, namely cynaroside and graveobioside A, in bell pepper leaves. The main objective of growing bell pepper plants, however, is the production of fruits of adequate quantity and quality for human nutrition. Considering the decline in plant biomass in response to stress conditions, it is very likely that the stressors applied would also lead to a reduction in fruit production. Severe salt stress, in particular, is known to be an important factor limiting crop productivity [45]. We have shown that the type of stressor has magnificent effects on both plant biomass and leaf secondary metabolite content. Other studies have proven that this also applies for different levels of abiotic stress [14,46]. The search for the best stressors and stress levels for the accumulation of secondary metabolites in plant leaves with negligible effects on fruit yield is a major future challenge for research in stress physiology. Several authors reported neutral or positive responses of product quality to mild stress [46]. For salt stress, several studies in the model-crop tomato reveal positive impacts of mild stress on fruit quality (e.g., antioxidant capacity and nutritional value) [47,48]. Low UV radiation reduces the antioxidative capacity and, therefore, the fruit quality of bell pepper fruits [49]. Additional UV radiation may help to overcome this problem and, at the same time, induce the production of valuable SM in the leaves. Cultivation of plants under mild water stress conditions can also enhance water use efficiency. To avoid any competition with food production, post-harvest treatment of leaves could be an appropriate measure to achieve high contents of promising metabolites [50,51]. These effects should also be taken into account when evaluating the value of production systems that are based on commercialization of both fruits and SMs in leaves of horticultural plants.

To enhance precision of non-invasive estimation of SMs in pepper leaves, future studies should consider hyperspectral sensors as well as chlorophyll fluorescence-based sensors, ideally a combination of both. Sensors covering the UV range are just entering the market and appear as a promising tool to access SMs in plants, as they cover absorption bands of flavones and other phenolic leaf compounds [52].

## 5. Conclusions

Both additional UV light and salt stress can enhance concentrations of the two SMs graveobioside A and cynaroside in bell pepper leaves. Highest concentrations were reached by combining both treatments. Stressed bell pepper leaves contain up to 30 mg graveobioside A and about 2 mg cynaroside per gram dry weight. While salt stress (100 mM NaCl) has a major negative impact on plant vegetative growth, UV stress (UVA 4–5 W m$^{-2}$; UVB 10–14 W m$^{-2}$; 3 h per day) has no significant impact on the fresh mass of the plants. The tendency of decreasing SM contents in leaves during

ontogenesis is outweighed by the stress treatments. Graveobioside A contents can be assessed with the multiparametric fluorescence sensor Multiplex. Reliable quantification of cynaroside is not possible with the non-invasive sensors used. If future experiments exclude major negative impacts on fruit quality, UV stress can be recommended as one tool to enhance valuable SMs in bell pepper leaves and potentially in vegetable leaves in general. A less-intense salt stress should also be considered in future experiments.

**Author Contributions:** Conceptualization, S.R.-S.; data analysis, J.E., N.S., P.K., J.-B.S.L., D.P., M.R., and L.S.; methodology, S.R.-S.; writing—original draft preparation, J.E.; writing—review and editing, J.E., S.R.-S., N.S., P.K., J.-B.S.L., D.P., M.R., and L.S. All authors have read and agreed to the published version of the manuscript.

**Funding:** This research was supported by the German Federal Ministry of Education and Research (grant number: 031B0361C).

**Acknowledgments:** The authors are grateful to Libeth Schwager for her support in laboratory analysis, and for plant cultivation by the staff members of the plant service team "Dienstleistungsplattform" of the University of Bonn. We thank Eduardo Fernandez for discussions and support in data visualization. We appreciate the support of Katharina Krah, Simone Klein, Miriam Brink, and Mark Schmutzler during the measurements. We are thankful for the great support by our project partners Manuel Lück, Anika Wiese-Klinkenberg, Julia J. Reimer, and Alexandra Wormit.

**Conflicts of Interest:** The authors declare no conflict of interest.

## Abbreviations

DATI    Days after treatment inception
HPLC    High performance liquid chromatography
ROS     Reactive oxygen species
SM      Secondary metabolite

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
