# Peer review of "Effect of UV Radiation and Salt Stress on the Accumulation of Economically Relevant Secondary Metabolites in Bell Pepper Plants"

_agronomy, doi:10.3390/agronomy10010142_

Round 1

Reviewer 1 Report

Dear author and editor,

in my opinion the manuscript "Effect of UV Radiation and Salt Stress on the Accumulation of Economically Relevant Secondary Metabolites in Bell Pepper Plants" can be considered for publication after minor revisions. In my opinion, the main constraint of the paper is the length of the trial: if plants were grown for their total cycle length and SMs were analyzed on both the final green biomass and fruits the manuscript would have had a remarkably higher impact. Nevertheless, I have found the manuscript very well written and informative, thus I would only suggest the authors to (better) explain the reasons behind the choice of testing plants for maximum 16 days after treatments introduction. Please find below my specific comments per section.

Abstract

Clear and concise. I have a remark only on the last sentence "High-UV stress is a promising tool for enriching plant leaves with valuable SM without major effects on plant biomass" which is too speculative, since the species cycle can last 3-4 months, according to different varieties, and you measured it on a much shorter period. As mentioned above, authors should justify the choice of the trial length and rephrase the results/conclusions accordingly.

Introduction

In general the chapter is ok. I suggest to better introduce the effects on salinity on the production of SMs (currently only lines 54-57), as it is done for UV effects.

line 66 (and throughout the manuscript) please remove the capital letter after ":"

Materials and methods

line 104: please explain why you did choose 100mM NaCl as salt stress concentration.

line 115: in this paragraph (plant harvest) please better specify the length of the trial: the harvest was performed 16 days after treatments introduction but how many days are to be counted before? 14 days of salt stress, whereas for UV treatment it is not specified.

Results

line 156-158: it is not written to which figure refers this sentence "Levels of SM in leaves of different ontogenetical stages are shown as an illustration of uneven distribution within the plants. SM contents decrease with leaf ontogenetical stage"

Discussion

line 223-225: "Stunted growth is a well described symptom of salt stress in plants [12,32], whereas plants reaction to UV-B exposure varies from growth reduction to enhancement, dependent on species, cultivar and stress level": I do not like very much this sentence, as also salt stress effects depends on species, cultivar and stress level. You aimed at enhancing SMs production through a moderate stress, but did you consider that perhaps 100mM NaCl was a too high concentration for the tested cultivars?

line 279-281: as mentioned already, the weakness of this manuscript is that the SMs have not been quantified on plants at the commercial maturity, that is the effective time of dealing with "bio-waste utilization". Moreover when you say that "Salt stress in particular is known to be an important factor limiting crop productivity" you are not making differences among severe and moderate salt stress: this is a part of your discussion that I suggest to enrich, as moderate salt stress is proven to enrich crop productivity (at least in terms of quality). Please have a look to the following publications in this respect:

Sgherri, C., Navari-Izzo, F., Pardossi, A., Soressi, G.P., Izzo, R., 2007. The influence of diluted seawater and ripening stage on the content of antioxidants in fruits of different tomato genotypes. J. Agric. Food Chem. 55, 2452–2458. https://doi.org/10.1021/jf0634451.

Sgherri, C., Kadlecova, Z., Pardossi, A., Navari-Izzo, F., Izzo, R., 2008. Irrigation with diluted seawater improves the nutritional value of cherry tomatoes. J. Agric. Food Chem. 56, 3391–3397. https://doi.org/10.1021/jf0733012.

Atzori, G., Mancuso, S., Masi, E., 2019a. Seawater potential use in soilless culture: a review. Sci. Hortic. 249, 199–207. https://doi.org/10.1016/j.scienta.2019.01.035.

Author Response

Dear Reviewer,

Thank you very much for your detailed review. We really appreciate your constructive critique and believe, that your comments helped us to improve our work. Please find detailed replies to your points below and in the updated manuscript (attached). As line numbers changed due to the adjustments, I´m referring to the “new” line numbers in my response. Your comments are displayed in italic, my responses in plain text.

Best regards,

Jan Ellenberger

Dear author and editor,

in my opinion the manuscript "Effect of UV Radiation and Salt Stress on the Accumulation of Economically Relevant Secondary Metabolites in Bell Pepper Plants" can be considered for publication after minor revisions. In my opinion, the main constraint of the paper is the length of the trial: if plants were grown for their total cycle length and SMs were analyzed on both the final green biomass and fruits the manuscript would have had a remarkably higher impact.

I agree, a next step should be to analyze the performance over a complete cycle and include fruits in our analyses. However, we wanted to proof that the methodology (kind of stress, level of stress, monitoring with our non-invasive tools) is appropriate for our purpose. We hope that upcoming prolonged and more complex experiments will produce results that have an even higher value for the scientific community.

 Nevertheless, I have found the manuscript very well written and informative, thus I would only suggest the authors to (better) explain the reasons behind the choice of testing plants for maximum 16 days after treatments introduction.

The rather short experimental period was still long enough to (a) see that the stressors effectively trigger accumulation of the SMs of interest and (b) the non-invasive devices allow us to some extend to quantify SM contents.
Expensive and potentially harmful UV-lamps are located in a closed growth chamber with limited space. It was therefore not feasible to reproduce the experimental setup for larger generative plants in our facilities. However, salt stress experiments on generative 6-months-old plants revealed that salt application only did not result in a comparably high increase of target metabolites.

Please find below my specific comments per section.

Abstract

Clear and concise. I have a remark only on the last sentence "High-UV stress is a promising tool for enriching plant leaves with valuable SM without major effects on plant biomass" which is too speculative, since the species cycle can last 3-4 months, according to different varieties, and you measured it on a much shorter period. As mentioned above, authors should justify the choice of the trial length and rephrase the results/conclusions accordingly.

We added the age of used plants in the abstract (l. 16) and softened the strong claim you refer to (lines 27 – 28).

We furthermore added the sentence “Effects on the performance of plants throughout a complete production cycle have to be evaluated in future trials.” In the abstract. (l. 29)

Also added the sentence “A less intense salt stress should also be considered in future experiments.” In the conclusions. (lines 348 – 349)

Introduction

In general the chapter is ok. I suggest to better introduce the effects on salinity on the production of SMs (currently only lines 54-57), as it is done for UV effects.

Thanks for that comment, you are right: the salinity-part was really short, especially when directly compared to the UV-part. I added an additional small paragraph at lines 57 – 60 and also added another study on salt response in bell pepper as a reference.

line 66 (and throughout the manuscript) please remove the capital letter after ":"

A German habit, I corrected that, thanks.

Materials and methods

line 104: please explain why you did choose 100mM NaCl as salt stress concentration.

Added “[…, ] since that concentration was shown to trigger higher total phenolic content in leaves of bell pepper seedlings in a previous study [26]. “ (ll. 118 -119)

line 115: in this paragraph (plant harvest) please better specify the length of the trial: the harvest was performed 16 days after treatments introduction but how many days are to be counted before? 14 days of salt stress, whereas for UV treatment it is not specified.

Added “[…] at a plant age of 48 days”. (l. 138)

Also corrected “14 days” to “16 days” regarding salt stress. (l. 117) and conducted minor changes to clarify plant ages and stress duration (lines 121 – 122)

Results

line 156-158: it is not written to which figure refers this sentence "Levels of SM in leaves of different ontogenetical stages are shown as an illustration of uneven distribution within the plants. SM contents decrease with leaf ontogenetical stage"

Added “(Fig. 1)” at the end of the sentence. (l. 183)

Discussion

line 223-225: "Stunted growth is a well described symptom of salt stress in plants [12,32], whereas plants reaction to UV-B exposure varies from growth reduction to enhancement, dependent on species, cultivar and stress level": I do not like very much this sentence, as also salt stress effects depends on species, cultivar and stress level. You aimed at enhancing SMs production through a moderate stress, but did you consider that perhaps 100mM NaCl was a too high concentration for the tested cultivars?line 279-281: as mentioned already, the weakness of this manuscript is that the SMs have not been quantified on plants at the commercial maturity, that is the effective time of dealing with "bio-waste utilization". Moreover when you say that "Salt stress in particular is known to be an important factor limiting crop productivity" you are not making differences among severe and moderate salt stress: this is a part of your discussion that I suggest to enrich, as moderate salt stress is proven to enrich crop productivity (at least in terms of quality). Please have a look to the following publications in this respect:

Thanks for the comments on the discussion section and the recommendations below. I changed the text in a way that it´s clear now that the level of stress does matter – as it does for UV exposure and pretty much in general. I also used the publications you recommended to enrich that section.

The changes are in particular:

Adding the sentence “If the applied salt concentration would have been lower, negative effects could probably have been avoided to a certain extent, as recently discussed in a review on the potential of seawater use in soilless culture [13].” (lines 253 - 256) Adding more precise info on stress level (lines 265 - 266) Adding the sentence “For salt stress, several studies in the model-crop tomato reveal positive impacts of mild stress on fruit quality (e.g. antioxidant capacity and nutritional value) [45,46].” (lines 322 – 324)

Sgherri, C., Navari-Izzo, F., Pardossi, A., Soressi, G.P., Izzo, R., 2007. The influence of diluted seawater and ripening stage on the content of antioxidants in fruits of different tomato genotypes. J. Agric. Food Chem. 55, 2452–2458. https://doi.org/10.1021/jf0634451.

Sgherri, C., Kadlecova, Z., Pardossi, A., Navari-Izzo, F., Izzo, R., 2008. Irrigation with diluted seawater improves the nutritional value of cherry tomatoes. J. Agric. Food Chem. 56, 3391–3397. https://doi.org/10.1021/jf0733012.

Atzori, G., Mancuso, S., Masi, E., 2019a. Seawater potential use in soilless culture: a review. Sci. Hortic. 249, 199–207. https://doi.org/10.1016/j.scienta.2019.01.035.

Reviewer 2 Report

I have two questions on this manuscript:

Why the specific selection of the two analysed metabolites

The determination of the spatial secondary metabolite contents (Table 3) is presented comparing all leaf vs leaf 4. I wonder why the authors have not compare young vs mature leaf since samples has been collected separating them.  

Author Response

Dear Reviewer,
Thank you for your questions regarding the manuscript. Your comments are displayed in italic, my answers in plain text. An updated version of the manuscript is attached.

Best regards,
Jan Ellenberger

I have two questions on this manuscript:
Why the specific selection of the two analysed metabolites

The two metabolites cynaroside and graveobioside A were chosen because of their relatively high monetary value and because previous experiments in the frame of the TaReCa project indicated that the concentrations of those metabolites are rather high in bell pepper. Furthermore, those specific SMs have medicinal applications that make them interesting substances for the pharmaceutical industry, as outlined from line 84 going.

The determination of the spatial secondary metabolite contents (Table 3) is presented comparing all leaf vs leaf 4. I wonder why the authors have not compare young vs mature leaf since samples has been collected separating them.

Thank you for your question.
I suppose you refer to table 2. This table contains the results of an ANOVA, calculated to analyze differences in the data presented in Fig. 4. We chose to present the comparison of the young “Leaf 4” vs all leaves, to point out the extraordinary strong reaction to stressors in young leaves, not only compared to oldest leaves, but compared to the average of the whole plant.

Reviewer 3 Report

Secondary metabolites are valuable from the perspective of medicinal purpose. This study demonstrated that abiotic stress such as salinity or UV irradiation can increase the contents of natural products in bell pepper plants. The overall study is interesting and well designed. The conclusion is clear and convincing. However, I do have some comments for the authors to consider:

line 18: "highest contents" should be changed to "highest contents of secondary metabolites".

Line 20: " aboveground" should be "above ground".

Line 22: "in" should be "into the".

Line 23: "over time" should be deleted.

line 36: "and legacy" should be deleted

line 36: "towards cascade use" should be "towards a cascade"

line 37: "pave the way for extracting" should be "pave the way for extraction and use..."

line 41: "gaining in importance" should be "gaining importance".

line 44-47: need reference

line 48: "but" should be deleted

line 47-49: these sentences have grammar issue

line 57: "induce senescence instead" should be "senesce"

Line 60-62: this paragraph does not seem to be complete. Please add more information.

line 79: "its positive effect ....drug cisplatin". What does this mean? Does the druy "ameliorate" the side effect"" If so, please reframe this sentence.

line 98: only one "all" should be kept

line 142: In Figure 1, please put the combined stress in the end of each graph; this is also the comment for other figures.

line 214: "thit" should be changed to "this".

line 228-231: 

It would be more informative to discuss about the trade-off. How much reduction in biomass vs how much SM increase?

Author Response

Dear Reviewer,

Thank you for your detailed and specific suggestions to improve the manuscript. We made most of the modifications you suggested and think that the message we wanted to transport to future readers is clearer than before. If not stated otherwise, we applied the changes as proposed. Detailed answers are presented below and an updated version of the manuscript is in the appendix and my line specifications refer to new line numbers in the updated manuscript (in the appendix).

Your text is presented in italics, my answers in plain text.

Best regards,

Jan Ellenberger

Secondary metabolites are valuable from the perspective of medicinal purpose. This study demonstrated that abiotic stress such as salinity or UV irradiation can increase the contents of natural products in bell pepper plants. The overall study is interesting and well designed. The conclusion is clear and convincing. However, I do have some comments for the authors to consider:

Thank you for the very specific suggestions that increase the quality of the manuscript.

line 18: "highest contents" should be changed to "highest contents of secondary metabolites".

Line 20: " aboveground" should be "above ground".

Line 22: "in" should be "into the".

Line 23: "over time" should be deleted.

line 36: "and legacy" should be deleted

What we intended to say is that the EU keeps changing legislation in ways that support the use of agricultural byproducts for all kinds of uses. Deleted it anyways, as it is not really needed in this context.

line 36: "towards cascade use" should be "towards a cascade"

line 37: "pave the way for extracting" should be "pave the way for extraction and use..."

line 41: "gaining in importance" should be "gaining importance".

line 44-47: need reference

Added a reference by Fini et al. in “Plant Signaling and Behavior” (DOI: 10.4161/psb.6.5.1506)

line 48: "but" should be deleted

line 47-49: these sentences have grammar issue

Split it into two sentences for more clarity.

line 57: "induce senescence instead" should be "senesce"

Line 60-62: this paragraph does not seem to be complete. Please add more information.

line 79: "its positive effect ....drug cisplatin". What does this mean? Does the druy "ameliorate" the side effect"" If so, please reframe this sentence.

Thank you for this comment, that really was a bad wording. Reframed that sentence(s):

“[…] it has the capability to prevent ROS-induced apoptosis in heart cells [22]. Cynaroside furthermore diminishes kidney injury as a side effect of cancer treatments with the chemotherapeutic drug cisplatin.” (lines 84 – 86)

line 98: only one "all" should be kept

line 142: In Figure 1, please put the combined stress in the end of each graph; this is also the comment for other figures.

Thank you for this comment, I see that there are good points for putting combined stress at the right side of each sub-graph. That would create a kind of continuous increase of stressors from left to right with 0, 1, 1, 2 stressors respectively.

We discussed this issue among the authors before handing in the manuscript and finally decided to put UV-stress at the end, since the reaction of the plants to UV stress is very different from the reaction to (a) single salt stress and (b) combined stress. That applies for the weights (Fig. 2) and also for the graveobioside a content (Fig. 1 C+D).line 214: "thit" should be changed to "this".

line 228-231: 

It would be more informative to discuss about the trade-off. How much reduction in biomass vs how much SM increase?

Yes, good point, we also came across that idea during the preparation of the manuscript but then dropped it at some point. We now added the sentence

“Finding the perfect trade-off between biomass and fruit yield loss on the one hand and SM increase on the other hand will be key to improve the production system.” (lines 262 – 264)

This tradeoff, however, cannot be made based on the results of this very study, since we lack information on fruit yield and the performance over a complete production cycle. We therefore state that future experiments should consider those points. When we (or someone else) does that in future, then even economic assessments of the trade-off are possible and should be considered.

Reviewer 4 Report

The authors have produced an excellent paper that provides valuable insights into stress induced secondary metabolites in greenhouse crops.   

consider “has been used” Please describe the plants more. You discuss greenhouse growing and growth chambers. Were they grown in the GH and then moved to the growth chamber? How long were they grown in the GH? How long were they in the chamber?

I think it would help the reader if you listed the factors being measured. I don’t see any info on dualex in M&M. Please be a little more descriptive. 4th mature leaf from the base of the plant? I think a description of the leaves is also needed in the foot notes for Figure 3.1.

168 to 170. This sentence doesn’t make sense to me. It sounds like a double negative. Did you mean to say that salt stressed plants were larger than combined but smaller than UV. I am ok with saying “

167 I think you can simplify this paragraph by saying that UV plants cv. Mavras and Stayer did not differ in DW from the control but were different that both the salt and the combined. I think either here or in the discussion the % increase/decrease in biomass should be reported. It might make strengthen the case for some of your comments in the discussion.

consider removing “under” “Thit” to “This”

309 and yield?

Author Response

Dear Reviewer,

Thank you for your valuable comments, your review helped us to improve the manuscript. Please find detailed answers below. Your original text is presented in italic, my answers in plain text. My line references apply to the updated version of the manuscript, which you can find in the appendix.

Best regards,

Jan Ellenberger

The authors have produced an excellent paper that provides valuable insights into stress induced secondary metabolites in greenhouse crops.   

consider “has been used” Please describe the plants more. You discuss greenhouse growing and growth chambers. Were they grown in the GH and then moved to the growth chamber? How long were they grown in the GH? How long were they in the chamber?
Thank you for pointing out these unclarities. We rewrote parts of the section 2.1 (Plant material and growth conditions) to address these issues (lines 103 – 110 and lines 118 - 122).

I think it would help the reader if you listed the factors being measured. I don’t see any info on dualex in M&M.
You are right, we somehow forgot that crucial point. Added a small paragraph and two references regarding Dualex: “Secondly, the transmittance-based fluorescence measurements were conducted with the Dualex sensor (Force-A, Orsay, France). The Dualex is a device with a leaf-clip, measurements are taken with virtually no distance to the leaf surface. The device is extensively described in the literature [29,30].” (lines 132-134).

Please be a little more descriptive. 4th mature leaf from the base of the plant?

Added “All leaf numbers are given as the number of true leaves, counted from the base of the plant.” In the methods (lines 144-145).

 I think a description of the leaves is also needed in the foot notes for Figure 3.1.
Changed the description in Fig 3 slightly to “[…] Color of points represents leaf age (Leaf 4, 10 and 12 as counted from the base, with darkest colors for youngest leaves).[…]” (line 217).

Also changed Fig. 1 description by adding “Colored boxplots represent leaf age - subgroups (Leaf 4, 10 and 12 as counted from the base, with darkest colors for youngest leaves).” (lines 170 – 171)

168 to 170. This sentence doesn’t make sense to me. It sounds like a double negative. Did you mean to say that salt stressed plants were larger than combined but smaller than UV. I am ok with saying “

Thank you for pointing at this sentence, it was wrong in the original manuscript. I changed the paragraph, also incorporating your below comment:

“UV stressed plants of both tested cultivars exhibited higher fresh and dry weights than plants under salt stress and combined stress conditions (Fig. 2). Observed mean fresh weight decreases in salt stress and combined stress plants compared to control and UV stress were in the magnitude of 50 % (Fig. 2, C+D). The mean dry weight tended to be higher for salt stressed plants as compared to plants under combined stress, but lower than the dry weights of plants experiencing UV stress or control conditions respectively. (Fig. 2, A+B)” (lines 192 – 197)

167 I think you can simplify this paragraph by saying that UV plants cv. Mavras and Stayer did not differ in DW from the control but were different that both the salt and the combined. I think either here or in the discussion the % increase/decrease in biomass should be reported. It might make strengthen the case for some of your comments in the discussion.

Rewrote this section in the results: "UV stressed plants of both tested cultivars exhibited higher fresh and dry weights than plants under salt stress and combined stress conditions (Fig. 2). Observed mean fresh weight decreases in salt stress and combined stress plants compared to control and UV stress were in the magnitude of 50 % (Fig. 2, C+D). The mean dry weight tended to be higher for salt stressed plants as compared to plants under combined stress, but lower than the dry weights of plants experiencing UV stress or control conditions respectively (Fig. 2, A+B)." (lines 195 - 200)

consider removing “under” “Thit” to “This”

Done.

309 and yield?

Plants used in this trial were in the vegetative stage. However, we added thoughts on yield to the discussion, e.g. in
lines 262 -264: “Finding the perfect trade-off between biomass and fruit yield loss on the one hand and SM increase on the other hand will be key to improve the production system.”
lines 319 – 321: “The search for the best stressors and stress levels for the accumulation of secondary metabolites in plant leaves with negligible effects on fruit yield is a major future challenge for research in stress physiology.”
